# Multiple Mobile Sinks for Quality of Service Improvement in Large-Scale Wireless Sensor Networks

**DOI:** 10.3390/s23208534

**Published:** 2023-10-18

**Authors:** Abdelbari Ben Yagouta, Bechir Ben Gouissem, Sami Mnasri, Mansoor Alghamdi, Malek Alrashidi, Majed Abdullah Alrowaily, Ibrahim Alkhazi, Rahma Gantassi, Salem Hasnaoui

**Affiliations:** 1Communication System Laboratory (Sys’Com), National Engineering School of Tunis (ENIT), University of Tunis El Manar (UTM), Tunis 1002, Tunisia; bechir.gouissem@enit.rnu.tn (B.B.G.); salem.hasnaoui@enit.rnu.tn (S.H.); 2Computer Science Department, Applied College, University of Tabuk, Tabuk 71491, Saudi Arabia; malghamdi@ut.edu.sa (M.A.); mqalrashidi@ut.edu.sa (M.A.); i.alkhazi@ut.edu.sa (I.A.); 3IRIT (RMESS), University of Toulouse II, 31058 Toulouse, France; 4Department of Computer Science, College of Computer and Information Sciences, Jouf University, Sakaka 72341, Saudi Arabia; malrowaily@ju.edu.sa; 5Department of Electrical Engineering, Chonnam National University, Gwangju 61186, Republic of Korea; rahmag@jnu.ac.kr

**Keywords:** cluster-based routing protocol, energy consumption, quality of service, multiple mobile sinks, large scale wireless sensor network

## Abstract

The involvement of wireless sensor networks in large-scale real-time applications is exponentially growing. These applications can range from hazardous area supervision to military applications. In such critical contexts, the simultaneous improvement of the quality of service and the network lifetime represents a big challenge. To meet these requirements, using multiple mobile sinks can be a key solution to accommodate the variations that may affect the network. Recent studies were based on predefined mobility models for sinks and relied on multi-hop routing techniques. Besides, most of these studies focused only on improving energy consumption without considering QoS metrics. In this paper, multiple mobile sinks with random mobile models are used to establish a tradeoff between power consumption and the quality of service. The simulation results show that using hierarchical data routing with random mobile sinks represents an efficient method to balance the distribution of the energy levels of nodes and to reduce the overall power consumption. Moreover, it is proven that the proposed routing methods allow for minimizing the latency of the transmitted data, increasing the reliability, and improving the throughput of the received data compared to recent works, which are based on predefined trajectories of mobile sinks and multi-hop architectures.

## 1. Introduction

WSN (Wireless Sensor Networks) is a special case of Ad hoc networks [1], broadly used in various applications such as environment monitoring, object tracking, military surveillance, traffic control, healthcare, etc. A WSN is a collection of large numbers of sensor nodes (SN) distributed over a geographic area to monitor certain phenomena. Each sensor node is limited in processing capability, wireless bandwidth, battery, and memory capacity. Mostly, it is difficult, even impossible, to recharge or change the battery, making energy consumption a significant constraint of WSNs lifetime [1,2].

WSNs have many advantages, and they are widely used due to their low cost, wireless communication capability, energy efficiency, and scalability, and they are suitable for Real-time monitoring applications.

The SNs can sense, process, and transmit data either via multi-hop transmission or directly to a base station (BS). The BS sends the collected data to a remote-control station through radio networks or satellite connections. WSNs have unique features like autonomy, self-organization, and Ad-hoc infrastructure, which makes them ideal for healthcare, smart cities, and environmental surveillance [2,3].

Since wireless communication requires significantly more power than other tasks, energy conservation is important while designing routing protocols for WSNs. The clustering approach is one of the best techniques for reducing the energy consumption of nodes. Therefore, instead of each node sending its collected data individually, first, sensor nodes organize themselves into clusters, and then an elected cluster head (CH) sends all aggregate data to the sink.

Clustering is used in WSNs for several important reasons, as it offers several benefits that contribute to the efficient and effective operation of these networks, such as:Energy Efficiency: As sensor nodes, WSNs are usually powered by batteries with energy resources clustering, which plays a role in evenly distributing energy-intensive tasks like data transmission and aggregation among the nodes. By assigning nodes as Cluster Heads (CHs) for collecting and aggregating data, non-CH nodes can conserve energy and extend the network lifespan by operating in low-power modes for longer periods.Reduced Communication Overhead: In clustered WSNs, the sensor nodes within a cluster typically transmit collected data to their respective CH. The CH, then. Forwards the data to the base station or sink node. This approach reduces communication distances within the network since data does not need to be transmitted to the base station. Consequently, reduced communication distances lead to energy consumption and alleviate network congestion.Scalability: With clustered WSNs, new nodes can easily join existing clusters as the network expands while CHs efficiently route data towards the Base Station (BS). This allows for network expansion without impacting its performance.Load Balancing: Cluster Heads are vital in distributing data collection tasks among sensor nodes within their cluster. This ensures that no single node becomes overwhelmed with the responsibilities of gathering data. This load-balancing technique plays a role in avoiding failures of nodes caused by excessive energy usage. Additionally, clustering enhances the fusion of data, allowing for aggregation at the CH level. As a result, redundant information collected by nodes is minimized, leading to the transmission of precise and concise data to the base station.Prolonged Network Lifetime: The combination of reduced energy consumption, efficient communication, and optimized data routing achieved through clustering significantly extends the overall network lifetime.

In large-scale WSNs, coverage is one of the most important QoS metrics, and it refers to how well the SNs in the network can monitor or sense the region of interest. Coverage directly impacts the ability of a WSN to fulfill its intended purpose, which could be environmental monitoring, surveillance, or any other sensitive application. In such a context, the battery replacement of large amounts of nodes is a labor-consuming work. Although the life of WSNs can be prolonged through energy-harvesting (EH) technology, it is necessary to design an energy-efficient routing protocol for energy harvesting, as an important part of nodes would be unavailable in the energy harvesting phase. In this phase, a certain number of unavailable nodes would cause a coverage hole, affecting the WSN’s monitoring function of the target environment. 

In [4], authors propose an adaptive hierarchical clustering-based routing protocol for EH-WSNs (HCEH-UC) to achieve uninterrupted coverage of the target region through the distributed adjustment of the data transmission. The proposal balances the energy consumption of nodes. Then, a distributed alternation of working modes is proposed to adaptively control the number of nodes in the energy-harvesting mode, which could lead to uninterrupted target coverage. The simulation results show that the proposed HCEH-UC protocol can prolong the maximal lifetime coverage of WSNs compared with the conventional routing protocol and achieve uninterrupted target coverage using energy-harvesting technology.

Despite this, numerous challenges such as Quality of Services (QoS), efficiency of used energy, mobility, and lifetime restrict the use of WSN. The QoS and energy consumption are relevant metrics used to assess the quality of paths in any designed routing protocol in WSNs.

The Quality of Service (QoS) is defined by the International Telecommunication Union regulations (ITU-T Supp. 9 of E.800 Series) [5] as the totality of characteristics of a telecommunications service that bear on its ability to satisfy stated and implied needs of the user of the service. Also, the QoE (Quality of Experience) is defined as the degree of delight or annoyance of the user of an application or service [6].

QoS in WSN refers to the ability of the network to provide certain guarantees regarding latency, throughput, packet loss, and reliability for different types of traffic. Since WSNs are typically deployed in harsh environments where resources are limited, providing QoS is a hard task to resolve. However, it is essential to meet the application requirements, such as monitoring critical infrastructure, conserving energy, and collecting data.

QoS is a very challenging subject, one of the big defis is how to guarantee QoS. In [7], a new method for QoE parameters prediction in an overall telecommunication system consisting of users and a telecommunication network, based on QoS indicators’ values prediction, are overviewed. The presented results show the advantages of the proposed overall model normalization techniques towards adequate prediction and presentation of QoE in conjunction with QoS in the overall telecommunication systems.

Besides, the small battery energy is a major constraint for WSNs. As these nodes are typically deployed in remote and inaccessible locations, recharging them is not feasible. Therefore, the energy resources of the sensors must be used efficiently to prolong the network’s lifespan [8,9]. If the network’s topology is not variable and the sink remains fixed, the energy distribution will be increasingly uneven over time. The network’s longevity is a crucial evaluation standard used to assess its performance, and it is typically measured by determining the period when the first node dies. Over the years, numerous routing protocols and algorithms have been suggested for energy-efficient WSNs, but many of these works suppose that the sink is static [10,11,12,13]. In routing protocols based on multi-hop communications, nodes close to the sink play a crucial role in transmitting data to other sensors. As a result, their energy resources tend to deplete faster, resulting in the hot spots issue [14].

Routing protocols using clustering help sensors sense and reassemble data from the environment and transmit it to the sink with minimum costs. By grouping nodes, clustering algorithms enhanced the performance of nodes and their ability to send data. In cluster-based routing protocols, even cluster heads (CHs) located far away from the sink are more likely to exhaust their battery reserves than those nearby since the needed hops for sending data increase with the square of the distance [15,16]. Node deaths can disrupt the network topology, reduce sensing coverage, and potentially result in a network partition, isolation of nodes, and loss of data. Additionally, in real-time WSN applications, such as military zone monitoring, enemy surveillance, natural disaster tracking (e.g., seismic activities), and exploration of inaccessible areas, stringent quality-of-service constraints are essential. These constraints include high data reliability, throughput network, low data delivery latency, and high communication efficiency, apart from efficient energy usage.

Clustering technology is crucial in reducing the consumed power by attributing sensors to clusters based on specific rules. The cluster features a set of CHs that act as relay nodes for other members within the group. Clustering simplifies the network topology and mitigates the need for sensor-sink communications. Moreover, CHs can leverage data fusion techniques to eliminate repetitive data, thereby lessening the CHs burden. A prominent example of a routing protocol which uses clustering is the “low-energy adaptive clustering algorithm” (LEACH). However, selecting CH in the LEACH is not optimal, and research work is needed to refine the protocol.

Moreover, mobile WSN (MWSN) is a novel variant of networks used in dynamic and mobile environments due to its capacity for self-configuration. In large WSNs, the network can be logically portioned into sub-networks. Each one has its mobile sink. Using mobile sinks is a highly effective approach for managing the imbalanced energy of WSNs. WSNs supporting mobile sinks typically deploy intelligent vehicles or robots to carry the sink, which can be moved freely around the sensing field. The implementation of mobile sinks was suggested and evaluated to address the imbalanced energy problem in WSNs [17,18,19,20,21,22,23].

Mobile sinks controlling region of interest (RoI) gather information from static sensors in one or multiple hops. A significant advantage of multiple mobile sinks is that they can distribute the communication and computation load across the network, reducing the burden on individual sensor nodes. This can significantly enhance the network lifetime by mitigating the effects of energy depletion in specific nodes. Moreover, multiple mobile sinks can help decrease data collection and transmission latency. By deploying sinks in different parts of the network, the time taken to collect and transmit data from distant nodes can be significantly reduced. However, there are some challenges in deploying mobile and multiple sinks. One such challenge is finding an optimal placement of sinks to cover the entire network efficiently. Also, the synchronization of mobile and multiple sinks can be complex. In conclusion, deploying multiple mobile sinks in a WSN is a promising approach for achieving better efficiency, network lifetime, and data collection, but it requires careful deployment and synchronization to realize its full benefits.

The current research aims to investigate how energy consumption and QoS metrics are enhanced by multiple mobile sinks that use a cluster-based routing protocol. We will examine four different mobility models on these metrics, focusing on identifying the number and cost of deploying mobile sinks. Our study aims to provide insights into how network performance can be enhanced while balancing its investment costs.

The next sections are as follows: Section 2 introduces previous works and compares them with our contribution. Section 3 presents the cluster-based routing protocols used in this study. Section 4 discusses the suggested sink mobility models. Section 5 highlights the findings of the simulations, along with analysis and discussions. Section 6 summarizes the paper and its perspectives.

## 2. Review of the Literature

This section discusses the studies utilizing stationary and movable WSN sinks to reduce power consumption and extend the network’s operating life. We will also discuss the QoS challenges in WSNs applications in the literature. 

In [24], a scheme for maximizing the longevity of WSNs utilizing a movable sink was suggested to manage the delays in delivering data. Each node has a range of tolerance for delay, within which it does not need to instantly transmit data when it is available. Instead, the node can keep data in storage for a while and transfer it at the appropriate time, i.e., when the mobile sink is at an optimal location to lengthen the network’s useful functioning duration.

Moving sink nodes is among the viable ways used to extend the lifetime of the network. As pointed out in [25], this technique can significantly improve network durability. In [26,27], the authors delve further into using numerous mobile sinks to enhance energy efficiency and network longevity. In another study [28], a joint optimization assessment to optimize the network lifetime using mobile sinks is performed by determining K-optimal trajectories and scheduling of sojourn time per position while abiding by the given constraints by sensors and mobile sinks.

Hence, mobility is a prevalent approach for mitigating hotspots’ issues and extending the lifetime of multi-hop WSN routing, as highlighted in [29]. Other studies, such as [30], highlight the impact of using mobile sinks on power usage and longevity by selecting optimal sink node numbers and parking positions. In [31], a network restructuring process is proposed by modifying the adjacent nodes of a sink to optimize the lifetime and balance the power usage among sinks. 

By ensuring that the total energy of the sink is below a specific threshold, only a set of selected nodes are connected, which enhances the network lifespan. Research in [32] highlights the benefits of using mobile sinks to prolong network life by randomly deploying nodes in a square area or pre-defined rectangular or hexagonal grids. The hexagonal grid deployment strategy is particularly effective since it maintains coverage and connectivity.

In the previously discussed studies, authors highlight only the issues of energy consumption with multi-hop routing, considering some fixed or mobile sinks to reduce energy and improve network lifetime. However, do not give importance to ever-increasing QoS criteria, especially in real-time constraint applications. Since such routing already suffers from an exhilarating node energy consumption and a huge data delivery delay due to the transfer of data between nodes until reaching the BS in multi-hop traveling.

Our study investigates a more critical problem in the real-time WSN context; we use multiple mobile sinks to improve energy consumption and QoS metrics. Existing cluster-based routing already guarantees better energy conservation and fast data delivery [15,16,17,18,19].

We are also trying to find the optimal number of mobile wells that maintain good power and QoS performance while considering the extra costs of mobile sinks deploying.

The weaknesses of the work based on multi-hop routing are mainly the border nodes ensuring routing data of their affiliated sensors. These border nodes quickly lose their energies and die, creating network partitioning and a huge loss of relevant data, especially in a military context or vital monitoring.

For that, we decided to work with hierarchical routing, which showed its performance concerning power usage and latency, representing the main weakness of mobility models relying on predetermined and non-adaptive trajectories.

This type of mobility lacks the flexibility of adaptation, especially in a variable, stochastic and unpredictable military context. Suppose a final node loses energy, dies, and goes out of service. In that case, the mobile sink continues its regular trajectory, and all the nodes that transmit their data through it become unable to deliver their data to the destination. Therefore, we easily fall into the phenomenon of black holes, and the network becomes partitioned, which is unacceptable in critical applications. While random mobility models do not follow a trajectory, remain flexible, and adapt quickly to any change of context. 

Recently, there has been a significant interest in the development of cluster-based and power-efficient mobile protocols for routing. One such protocol, the “Energy-efficient Cluster-based Dynamic Routing Algorithm” (ECDRA) [33], involves deploying a mobile sink attached to a sensor that rotates circularly to dynamically change the topology of the network in response to the sink’s position. However, LEACH [14] is a well-known hierarchical routing protocol differentiating CHs from normal nodes (ONs). ONs transfer their data to the appropriate CHs, which collectively transmit the data to BS. While LEACH is more effective than classical routing protocols at increasing network lifetime, the random selection of CHs can result in uneven distribution and flow between the BS and the CHs, leading to higher energy consumption.

In [34], a framework that enhances energy efficiency and the QoS for WSN is presented. The introduced hybrid technique utilizes a fitness function that considers key performance indicators like the number of neighbors, the set of sensors for each cluster, and how long each node remains the CH. This fitness is integrated with a probability threshold function to influence the procedure of selecting CHs. Compared to previous homogeneous protocols such as LEACH, the proposed method maintains optimal CH selection more stably throughout network operation. Furthermore, compared with heterogeneous protocols like “Developed Distributive Energy-Efficient Clustering and Enhanced Developed Distributive Energy-Efficient Clustering”, the proposed protocol displays superior performance regarding the WSN lifetime, power usage, and throughput. However, this suggested routing algorithm should offer more privacy and security features. Moreover, to prove its consistency, this paradigm should be tested in a real-world context. 

The authors of [35] introduced an evaluation technique to compare the “Secure Mobile Sink Node Location Dynamic Routing Protocol” (SMSNDRP) with another algorithm named “routing protocol with K-means for forming Data Gathered Path” (KM-DGP). The application of these two algorithms was on networks with Mobile Sinks of various sizes. QoS and power usage are used to assess the quality of routes and energy consumption patterns of both routing protocols on small (with single and multiple mobile Sinks) and large networks. The proposed evaluation technique is implemented on NS3 using five different scenarios. The findings suggest that compared with KM-DGP, SMSNDRP shows improved network energy consumption on small, single networks. In contrast, for larger networks with sixteen mobile Sink nodes or more, KM-DGP displays comparatively better network energy consumption than SMSNDRP with four mobile sink nodes.

The study in [36] introduces a new high-performance communication protocol for routing packets using multiple mobile nodes. The protocol relies on four main features: assessment of packet delays, independent control of link quality and choosing active neighbors of the nodes. Simulation studies on this protocol show that the latter improves the packet forwarding rates, reduces power usage, and shortens average delays.

The authors in [37] presented a clustering paradigm for MWSN. Their technique involves introducing super cluster heads (SCH) that are static and efficient sensors within the MWSN to gather CH data from CHs. Combining SCHs with the “Minimum Transmission Energy” (MTE) protocol reduces the distance required to transfer data from CH to BS, ultimately improving energy efficiency. Under this approach, data is first transmitted from CH to SCH and then forwarded to BS. This new technique promises to enhance the network performance further.

Another study in [38] introduces a new energy-efficient routing system that employs clustering and sink mobility techniques. The authors propose a two-step approach that involves classifying the region of interest (RoI) into sectors and selecting a CH for each sector based on the weight of each node member. Afterwards, each member calculates the power usage of numerous routing paths and selects the most energy-efficient option. Finally, CHs are linked in a chain via a greedy strategy for inter-cluster connectivity. The findings show, as demonstrated through simulations that this new routing strategy is better than similar approaches, like “Cluster-Chain Mobile Agent Routing” (CCMAR) and “Energy-efficient Cluster-based Dynamic Routing Algorithm” (ECDRA).

According to a recent study [39], an auto-schedule routing algorithm relying on IoT connections was introduced to enhance the power usage of Software-defined networking (SDN) controlled embedded networks. The algorithm starts by constructing the “Neighbor Distance Discovery Protocol”, which identifies the “minimum depletion path” by locating the closest node to the BS. Next, the algorithm executes the “Multipath Cooperative Self-Scheduling Protocol” to establish a non-traffic route. Additionally, the algorithm involves the routing communications of each IoT object in building the routing medium. It computes the average packet loss rate, node response rate, energy consumption, sensor absorption rate, and transmission delay. Finally, the algorithm employs the “Lifetime Duty Cycled Energy Efficient Protocol” to determine the network threshold latency and energy limits.

The research discussed in [40] explores the latest routing algorithms used in sensor networks and proposes strategies for their development. This study highlights recent advances in the strategy used to reduce the energy required for information transmission. One key concern for IoT, which has gained much attention, is the energy requirements to extend the lifespan of IoT networks. One of the approaches that has gained traction is the design of routing protocols that minimize energy consumption during data transmission.

In recent studies, optimization paradigms have been utilized to address the energy issues in WSNs by means of an energy-efficient multi-objective criterion as follows:

The proposed clustering and optimization-based routing approach in [41] is used to improve the power efficiency and prolong the lifespan. The selection of CH is achieved in parallel with the minimization of power usage, which effectively reduces dead sensor nodes. The use of the “Sailfish optimizer algorithm” for optimal path selection also enhances the energy efficiency of data transmission between CH and BS. However, the study does not consider the node mobility in the proposed approach. In WSNs, nodes can move frequently due to environmental conditions or other factors. Hence, the network’s topology varies, which may affect the performance of routing algorithms. Future research could address this limitation by incorporating mobility models into the proposed approach to improve its adaptability to dynamic network conditions.

Moreover, some studies have attempted to enhance the network power usage and its lifespan via various optimization algorithms such as the PSO algorithm [42], bio-inspired ant colony [43], etc. Another research [44] introduced a hybrid ACO-PSO routing paradigm that employs mobile sinks to reduce overall power usage.

Hence, the research gap can be summarized as follows:

Despite the potential benefits of using multiple mobile sinks in WSNs, research gaps should be addressed in this domain. One of the significant gaps is the establishment of performant and robust routing paradigms for multiple mobile sinks since routing protocols determine the efficiency of WSNs. The challenge of multiple mobile sinks is to design a routing protocol to handle the changing positions of sinks and ensure efficient data delivery. Current routing protocols used for multiple mobile sinks are based on centralized approaches, which can lead to scalability issues and network congestion. Another research gap is related to the synchronization of mobile multiple sinks. When multiple sinks move in the RoI, it can be challenging to ensure that they are synchronized in terms of their locations and data collection schedules. Synchronization is essential to avoid collisions and ensure efficient data collection.

Furthermore, the deployment methodology of multiple mobile sinks is another area where research is needed. Identifying an optimal number of mobile sinks, their placement, and their trajectories requires sophisticated algorithms and optimization techniques. Overall, the research gaps regarding using multiple mobile sinks in WSNs include establishing scalable and high-performance routing protocols, synchronization techniques, and optimal deployment methods that can enable efficient and reliable data collection in large-scale WSNs. 

QoS in WSNs has been an interesting research topic in recent years. Many WSN real-time-based applications require the support of QoS. However, the development of sensor networks needs to consider various factors such as fault Tolerance, resource allocation, adaptive routing, data reliability, Real-time communication, scalability, and energy efficiency [45,46].

Addressing these QoS challenges in WSNs often involves a combination of hardware and software solutions, including efficient protocols, energy-efficient algorithms, and adaptive strategies suitable to the specific application requirements. Researchers continue to develop innovative approaches to overcome these challenges and enhance the performance of WSNs in various domains [47,48].

Data aggregation is a method to effectively reduce the data transmission volume and improve network lifetime. However, the data waiting for processing in the queue are subject to an extra delay. In this paper [47], the authors propose an Adaptive Aggregation Routing (AAR) scheme to avoid this problem by dynamically changing the forwarding node according to the length of the data queue and balancing the aggregating and data-sending load. Simulation results demonstrate that compared with the existing schemes, the proposed scheme reduces the delay by 14.91%, improves the lifetime by 30.91%, and increases energy efficiency by 76.40%.

Coverage is a fundamental QoS metric in WSNs that assesses the ability of the network to adequately monitor a target area. It involves careful node deployment sensing range configuration and may require adaptation strategies to maintain coverage over time. In [48], authors propose an energy-efficient clustering routing protocol based on a high-QoS node deployment with an inter-cluster routing mechanism (EECRP-HQSND-ICRM) in WSNs. The new protocol introduces a node deployment strategy based on twofold coverage. The proposed strategy divides the monitoring area into four small areas centered on the base station (BS), and the CHs are selected in the respective cells to satisfy the uniformity of the CHs distribution. The simulation results show that, compared with the general node deployment strategies, the deployment strategy of the proposed protocol has higher information integrity and validity and lower redundancy.

One of the important challenges is the uncertainty of the service of requests. Recently, intuitionistic fuzzy estimations of the QoS have been proposed, such as in this work [49], where three intuitionistic fuzzy characterizations of virtual service devices are specified: intuitionistic fuzzy traffic estimation, intuitionistic fuzzy flow estimation and intuitionistic fuzzy estimation about probability. Six intuitionistic fuzzy estimations of the uncertainty of comprise service devices are proposed. The proposed uncertainty estimations allow for the definition of new Quality of Service (QoS) indicators. They can determine the quality-of-service compositions across a wide range of service systems.

## 3. Methodology

### 3.1. Simulation Setup

The current research aims to assess the sensors using cluster-based routing protocols. The evaluation will investigate throughput, reliability, packet latency time, and energy consumption with four mobility models. The study compares the results of different sink positions, ranging from one to eight static and mobile sinks. To ensure accuracy and reliability, the simulation will be repeated 100 times for each scenario in different topologies. The Castalia/OMNET++ simulator [50] will be utilized for the simulation process. The “Throughput Test” application is implemented and used for this purpose.

Castalia is a discrete-event simulator specifically designed for WSNs and is built on the OMNeT++ simulation framework. To evaluate the energy consumption and the QoS in a large-scale WSN using Castalia, we follow these steps to set up the simulation, define parameters, and select appropriate metrics:

#### 3.1.1. Step 1: Install Castalia

Download and install the Castalia simulator v3.2 and OMNeT++ framework v5.0 according to the installation instructions provided on the Castalia website [50].

#### 3.1.2. Step 2: Create the Simulation Scenario

Define the geographical area or environment where the WSN will be deployed.Define the number and initial positions of SNs and the static sink in the network.Define the random or deterministic deployment strategy.Define the mobility patterns of sinks.

#### 3.1.3. Step 3: Configure Simulation Parameters

Edit the Castalia configuration file (Config.ini) for each simulation scenario shown in Appendix A.Configure various parameters, including but not limited to:
○Communication protocols (MAC and routing protocols).○Radio models and channel characteristics.○Node properties (battery capacity, transmission power, data rate).○Simulation time and warm-up period.○QoS-related parameters like latency, packet delivery ratio, and throughput requirements.○Energy models.


#### 3.1.4. Step 4: Define QoS Metrics

Select the specific QoS metrics you want to evaluate based on our research goals. Common QoS metrics in WSN simulations include Packet Delivery Ratio (Reliability), End-to-End Delay (Latency), Throughput, Network Lifetime and Coverage.

#### 3.1.5. Step 5: Run the Simulation

Build and run the simulation using the OMNeT++ IDE or command-line tools as per the Castalia documentation [50].Monitor and collect simulation results, which include the QoS metrics you defined in step 4.

#### 3.1.6. Step 6: Analyze and Interpret Results

Use the collected data to analyze the QoS performance of the WSN.Generate graphs, plots, and statistics to visualize and interpret the results.Draw conclusions based on the evaluation of QoS metrics and how they relate to our research objectives.

#### 3.1.7. Step 7: Iterate and Refine

Depending on our findings, we repeat and refine the simulation to further investigate or optimize QoS in our WSN.

### 3.2. Cluster-Based Routing Protocols

Many recent articles have treated the impact of sink mobility in the WSN with multi-hop routing mechanisms [15,16]. In such a type of routing, the nodes closest to the sink dissipate their energies rapidly since they retransmit the collected data from distinct sensors to the sink, which divides the network, isolates the sink, and creates energy holes.

The use of mobile sinks has considerably alleviated these concerns in terms of reliability, throughput, and consumed energy. However, the data delivery delay was still modest due to the accumulated delay for each hop. Furthermore, this routing technique is not suitable for larger networks since the required number of hops is influenced by the number of deployed nodes. In such case, other than the number of hops, the delivery delay increases, and interferences between packets also increases, which rapidly and significantly degrades the throughput [15,16].

The cluster-based routing protocols have proven good energy conservation results and low data delivery latency [11,12,13,14]. For these reasons, in this study, we investigate the effect of using multiple mobile sinks with a cluster routing paradigm in large-scale WSNs. We will introduce the technique of such a routing protocol with the LEACH [14], P-LEACH [51], and EA-CRP [52]. The Table 1 provides a brief comparison between the key features of the three routing protocols to be studied.

### 3.3. Sink Mobility Patterns

To ensure unbiased results towards any mobility model, the authors of this study chose to compare statics sinks outside the RoI with four random mobility models: Random WayPoint Mobility Model (RWP) [53,54]: A model that includes pause times between changes in directions and speed.Random Walk Mobility Model (RW) [53,54]: A simple mobility model based on random directions and speeds.Random Direction Mobility Model (RD) [53,54]: A model that forces MNs to travel to the edge of the simulation area before changing direction and speed.Gauss Markov Mobility Model (GM) [53,54]: A memory model that uses one tuning parameter to vary the degree of randomness in the mobility pattern.

The basis for selecting these models for evaluating network Quality of Service (QoS) metrics is that they support unpredictable and random changes, like real-time scenarios. Subsequently, we will delve into how each of these models operates. The functioning of each mobility model is described in the article [21].

Figure 1a shows an example of a traveling path of a sink, which begins in the center of the RoI, using the RW Mobility Model. At each point, the sink randomly chooses a direction between 0 and 2 and a speed between 0 and 10m/s. At every 60, the sink changes direction and speed. This Model is a memoryless mobility pattern because it retains no knowledge concerning its past locations and speed values [14].

Figure 1b shows an example of a traveling path of a sink, which begins in the center of the RoI, using the RWP Mobility Model. The movement pattern of a sink using the RWP Mobility Model is similar to the RW Mobility Model if pause time is zero and [min-speed, max-speed] = [speed-min, speed-max].

Figure 1c shows an example path of a sink, which begins in the center of the RoI, using the RD Mobility Model. In this model, the sink chooses a random direction to travel, similar to the RW Mobility Model. The sink then travels to the border of the simulation area in that direction. Once the RoI boundary is reached (represented by dots in the figure), the sink pauses for a specified time, chooses another angular direction (between 0 and 180 degrees) and continues the process. 

Figure 1d illustrates an example traveling pattern of a sink using the GM Mobility Model; the sink begins its movement in the center of the RoI and moves for 1000 s. The Gauss–Markov Mobility Model was designed to adapt to different levels of randomness via one tuning parameter. Initially, the sink is assigned a current speed and direction. At fixed intervals, n, movement occurs by updating the speed and direction. Specifically, the value of speed and direction at the nth instance is calculated based on the value of speed and direction at the n-1 instance and a random variable. This model can eliminate the sudden stops and sharp turns encountered in the RW Mobility Model by allowing past velocities (and directions) to influence future velocities (and directions).

## 4. Results and Discussion

### 4.1. Simulation Scenarios and Evaluation

In WSNs with cluster-based routing [55,56,57], the shorter the distance separating the sink from the CH is, the more the power is conserved, and the lower the packet collection latency is. Using numerous sinks to replace a single sink can significantly decrease these distances. Using multiple sinks, every cluster head can communicate with the nearest sink. Therefore, it is possible to enhance the QoS performances by deploying multiple sinks [58] or relay nodes [59,60] to gather sub-regional data. This technique has been proven effective in reducing distances and improving overall performance. Hence, it is a preferred solution for achieving a better quality of service performance.

As a result, the primary sensor network is partitioned into smaller networks with a low diameter. These sub-networks consist of sensors and a static or mobile sink, forming a cluster. Cluster heads transfer information to the respective sink of the corresponding sub-region.

The study’s initial scenario will focus on a simulation of a field measuring 400 m × 400 m, hosting eight hundred nodes with random positions. This used static sink is situated beyond the RoI. Further, the simulation will be repeated four times using a mobile sink, which follows one of the four designated mobility models in Figure 2a. Another scenario implies the same RoI and number of nodes, but the sensing field is split into two 400 m × 200 m sections.

Each half will be controlled, once with a static sink positioned beyond the supervised RoI, and four times with a mobile sink that follows one of the four mobility models, so we will have two fixed sinks and two mobile sinks, as illustrated in Figure 2b in the third scenario, it is the same principle as the second scenario except that we divide the main field into four subfields sized of 200 m × 100 m. In this case, we will have four fixed and four mobile sinks, as illustrated in Figure 2c. Finally, for the fourth scenario, we also keep the same principle as the other scenarios, but this time, we divide the initial field into eight subfields of 200 m × 100 m. In this case, we will have eight static and eight mobile sinks, as shown in Figure 2d.

The assessment analysis of the introduced system relies on various assumptions, including the supposed stationary state of all deployed sensor nodes, coupled with their location awareness. The utilization of a cluster-based routing protocol has also been considered, where only CHs are authorized to transfer gathered information to the designated sink. The latter changes its position through the network following a designated mobility model that facilitates data collection from the respective cluster heads. Each sensor node uniformly issues a standard amount of data per unit of time (i.e., one packet per second) with an equivalent data length of 100 bytes. 

The energy of transmission of each sensor is adapted to the adjacent nodes’ relative distances. The mobile sink is supposed to hold enough power reserves to communicate and relocate at any point within the network. The static base stations in each scenario are positioned 20 m beyond the field margins. According to the operating principles of the three hierarchical protocols described above, an election phase is planned each period (i.e., 20 s for LEACH) to choose a new CH.

Additionally, all sensor nodes have identical communication capacity and computing resources. Table 2 highlights the relevant, considered parameters.

The metrics chosen for evaluation are as follows:Energy consumption (The consumed energy by all the sensor nodes)Throughput (Total data collected by sink)Data delivery rate (Reliability)Delay or Packet Latency

To achieve accurate simulation results, we will use four random mobility models of the sink (GM, RW, RWP and RD) and a static model (a static sink located outside the region of interest; Fixed Sink) with three cluster-based routing protocols (LEACH, P-LEACH, and EA-CRP).

### 4.2. Energy Consumption Evaluation

In the WSN, one can never talk about network performance evaluation without studying the major concern of energy consumption. For that, in this simulation scenario, we compared the energy consumed by all sensor nodes (Network Energy) by varying the number of mobile and static sinks that monitor the network to study the impact of using multiple mobile sinks on energy consumption in large-scale WSNs (LS-WS). 

The simulation results illustrated in Figure 3a show that the use of a single mobile sink, regardless of the model, decreases the average energy consumed compared to a fixed sink by −12.5% for RW up to −19% for RD, and the best result obtained with the use of the RD mobility model and the P-LEACH routing protocol with a power gain of −20%.

The simulation results illustrated in the other figures show that the use of multiple mobile sinks decreases the average energy consumption compared to a single static sink, as illustrated in Figure 3a:By using two mobile sinks (Figure 3b) from −25.6% for RW up to −31.3% for RD and the best result obtained with the RD model and the P-LEACH routing protocol −36.1%.By using four mobile sinks (Figure 3c) from −44% for RW to −48.3% for RD, the best result was obtained with the RD model and the P-LEACH routing protocol −57.5%.Using eight mobile sinks (Figure 3d) from −48% for RW to −52% for RD, the best result was obtained with the RD model and the P-LEACH routing protocol −59.5%.

It can be drawn from the previous results that the best mobility model is RD, which offers better energy conservation, around 60% less, with eight mobile sinks, compared to a single fixed sink. More precisely, the sink with RD mobility model moves towards the clusters of sensor nodes to reduce the distance between the sink and the CH and consequently reduce the transmission energy consumption.

On the other hand, by comparing the different energy scenarios, we notice that using four mobile sinks gives results very close to that of eight mobile sinks. 

So, we can conclude that in terms of profitability, the use of four mobile sinks establishes a good compromise between energy conservation and investment cost (budget of mobile sinks), and we can obtain better energy conservation of 57.5% less with only four mobile sinks using the RD model and the P-LEACH routing protocol. Since consumed power is expected to increase exponentially as the communication range increases, utilizing a shorter transmission distance can significantly optimize the power consumed by mobile sinks. This implies that the power conservation of the network will be higher, and its lifetime will be extended as the subnet area becomes smaller. Nonetheless, it will incur a higher cost of deploying adequate mobile sinks to cover the area.

### 4.3. Throughput Evaluation

When deploying WSN applications with service quality constraints, the amount of collected data by the sink becomes important to consider.

To address this, in the second phase of the assessment, we analyzed the packets received by the sink (throughput) of the network with multiple mobile and static sinks.

In this simulation scenario, we compared the amount of data collected (Throughput) by varying the number of mobile and static sinks monitoring the network to study the impact of multiple mobile sinks on the throughput in the LS-WSNs. 

The simulation results illustrated in Figure 4a show that the use of a single mobile sink, whatever the model, increases the average flow rate compared to a fixed sink by +8.6% for RWP up to +14% for RD, and the best result obtained with the RD model and the EA-CRP routing protocol with a throughput gain of +15.5%. 

The simulation results illustrated in the other figures show that the use of multiple mobile sinks increases the average flow compared to that of a single static sink, as illustrated in Figure 4a respectively:Using two mobile sinks (Figure 4b) from +25% for RWP up to +31% for RD, the best result is obtained with the RD model and the EA-CRP routing protocol + 38%.Using four mobile sinks (Figure 4c) from +49% for RWP to +57% for RD, the best result is obtained with the RD model and the EA-CRP routing protocol + 73%.By using eight mobile sinks (Figure 4d) from 60.5% for RWP to 68.5% for RD, the best result is obtained with the RD model and the EA-CRP routing protocol +90.2%.

It can be deduced from the previous results that the best mobility model is RD, which offers almost double the throughput compared to a single fixed, using eight mobile sinks.

More precisely, the RD moves in the network by realistically approaching the clusters of SN to reduce the distance between the sink and the CH, consequently reducing the phenomena of collisions and packet retransmission and increasing the throughput.

On the other hand, by comparing the flow scenarios, we note that the use of four mobile sinks gives results very close to that of eight mobile sinks. So, we can conclude that in terms of profitability, the use of four mobile sinks establishes a good compromise between the flow rate and the investment cost, and we can obtain a better throughput of 73% more with only four mobile sinks by using RD mobility model and EA-CRP routing protocol.

### 4.4. Reliability Evaluation

In this simulation scenario, we compared the network’s reliability by varying the number of mobile and static sinks monitoring the network to study the impact of using multiple mobile sinks on reliability in LS-WSN. 

The simulation results illustrated in Figure 5a show that the use of a single mobile sink, whatever the model, increases the average reliability compared to a fixed sink from +17% for RWP up to + 20% for RD, and the best result is obtained with the RD model and the EA-CRP routing protocol with a gain of +22%.

The simulation results illustrated in the other figures show that the use of multiple mobile sinks increases the average reliability of the network compared to that of a single static sink (Figure 5a):By using two mobile sinks (Figure 5b) from +43% for RWP up to +52% for RD, the best result is obtained with the RD model and the EA-CRP routing protocol + 65%.By using four mobile sinks (Figure 5c) from +73% for RWK up to +88% for RD, the best result is obtained with the RD model and the EA-CRP routing protocol + 105%.By using eight mobile sinks (Figure 5d) from 91.5% for RWK up to 110% for RD, the best result is obtained with the RD model and the EA-CRP routing protocol + 140%.

It can be deduced from the previous results that the best mobility model is RD, which offers more than twice more throughput compared to a single fixed sink for the whole network, using eight mobile sinks. More precisely, the RD moves in the network by approaching in a realistic way towards the SN clusters to reduce the distance between the sink and the CH and consequently reduce the phenomena of collisions and packet retransmission, consequently increasing the network reliability, especially with the EA-CRP routing protocol which uses the combination of two routing techniques, clustering, and multi-hop.

On the other hand, by comparing the flow scenarios, we note that the use of four mobile sinks gives results very close to that of eight mobile sinks. So, we can conclude that in terms of profitability, the use of four mobile sinks establishes a good tradeoff between the network reliability and the cost of investment, and we can obtain better reliability of +105% more with only four mobile sinks by using the RD mobility model and the EA-CRP routing protocol.

### 4.5. Packets Latency Time (End-to-End Delay) Evaluation

In this simulation scenario, we compared packet latency (the percentage of packets that arrive at the sink with less than 1ms delay) by varying the number of mobile and static sinks monitoring the network to study the impact of the use of multiple mobile sinks on packet latency in LS-WSNs.

The simulation results illustrated in Figure 6a show that the use of a mobile sink, whatever the model, increases the mean percentage of fast packets (packets with delay less then 1ms) compared to that of a single fixed sink from +8% for RWP up to +28% for RD, and the best result is obtained with the RD model and the LEACH routing protocol +73%.

The simulation results shown in the other figures show that the use of multiple mobile sinks increases the mean percentage of fast packets compared to the use of a single static sink (Figure 6a):By using two mobile sinks (Figure 6b) from +21% for RWP up to +43% for RD, the best result is obtained with the RD model and the LEACH routing protocol +99.5%.By using four mobile sinks (Figure 6c) from +43% for RWP up to +69% for RD, the best result is obtained with the RD model and the LEACH routing protocol +153%.Using eight mobile sinks (Figure 6d) from 51% for RWP to 78% for RD, the best result is obtained with the RD model and the LEACH routing protocol +175%.

We can draw from the previous results that the best mobility model is RD, which offers almost triple results and 2.73 times faster packets compared to a single fixed sink by using eight mobile sinks. Therefore, we conclude that using multiple mobile sinks enhances the packet latency, particularly in smaller areas.

More precisely, the RD moves in the network by approaching them in a realistic way towards the SN clusters to avoid packet retransmission, decreases the distance between the sink and the CH, and consequently increases the number of packets that reach sinks with low delays due to the single hop routing technique used by LEACH.

On the other hand, by comparing the latency scenarios, we note that the use of four mobile sinks gives results very close to that of eight mobile sinks. So, we can conclude that in terms of profitability, using four mobile sinks establishes a good tradeoff between the latency time and the cost of multiple mobile sinks. So, we can obtain a better latency of +153% for fast packets with only four mobile sinks by using the RD model and the LEACH routing protocol.

In conclusion, when the set of static or mobile sinks increases their proximity to sensor nodes, fewer nodes are associated with each sink. This results in the reduction of the packet interference and the un-saturation of the sink buffer. This avoids packet retransmission and subsequently improves the number of packets that hold their sinks without additive delay.

Comparing the effect of static and mobile sinks, we can deduce that mobile sinks offer gains in energy conservation of −59.5%, throughput of +90.2%, reliability of +140% and latency of 175% with eight mobile sinks with the best RD the best mobility model for our scenarios. The use of eight mobile sinks offers a non-significant gain compared to the use of four mobile sinks. However, supplementary expenses were incurred due to the use of a high number of mobile sinks. So, we can just be satisfied with four mobile sinks with the RD mobility model, seeing the high costs of eight mobile sinks. Otherwise, it will be a waste.

## 5. Limitations and Potential Solutions

Conducting a simulation study to improve QoS in large-scale WSNs using multiple mobile sinks and random mobility models is a valuable approach. However, like any research methodology, it has limitations. The Table 3 below resumes the approach and the potential solutions or areas for future research:

## 6. Conclusions

This paper investigated the impact of using multiple mobile sinks on network energy efficiency and QoS metrics using a cluster-based routing approach and random mobility patterns.

More specifically, this type of protocol uses hierarchical routing, which offers good energy conservation and latency time results. Moreover, random mobility models run through the network in the context of a simulated battlefield observation where fast and unexpected events reoccur by getting closer to CHs to reduce power consumption during the transmission phase, reduce delays during data collection and increase the number of packets collected.

The simulation results obtained demonstrate that the Random Direction mobility model with four sinks has a significant impact on power consumption and QoS metrics. In particular, the EA-CRP and P-LEACH protocols achieve a significant improvement in terms of energy consumption, throughput, and reliability, while the latency time gains better with the LEACH protocol.

In addition, the simulation results show that RD is more suitable for LSWSN because it maintains good performance in terms of power consumption and all the QoS criteria studied for the large supervised RoI and that one can optimize the number of mobile sinks according to the real-time constraints of the RCSF application and the allocated budget.

After an in-depth discussion of the current approach limits regarding certain uncertainties in the operation of the mobility models used. We planned for future work to first, investigate the use of mobility traces collected from real-world deployments to create more accurate random mobility models. Secondly, consider a hierarchical routing protocol with random mobility awareness of the sinks to reduce the number of mobile sinks and associated costs while respecting QoS. Finally, optimize the mobility models to ensure they accurately reflect the movement of sinks in realistic WSN applications with QoS constraints. The proposed model may also be applied to numerous real-world engineering problems [61,62].

## Figures and Tables

**Figure 1 sensors-23-08534-f001:**
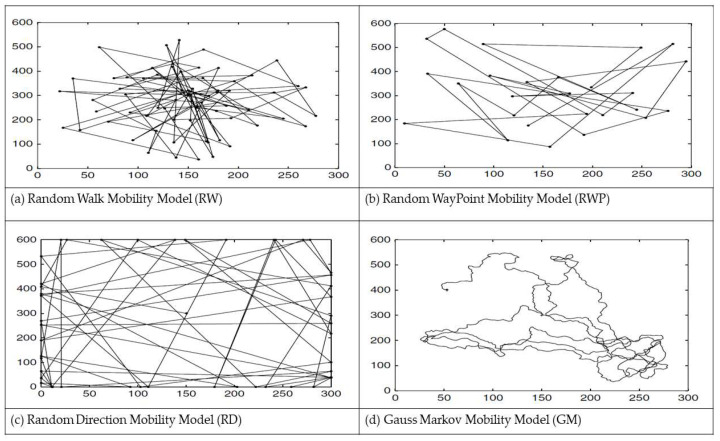
Traveling patterns of a Mobile Sink using (**a**) the Random Walk MM (RW), (**b**) the Random WayPoint MM (RWP), (**c**) the Random Direction MM (RD) and (**d**) the Gauss Markov MM (GM).

**Figure 2 sensors-23-08534-f002:**
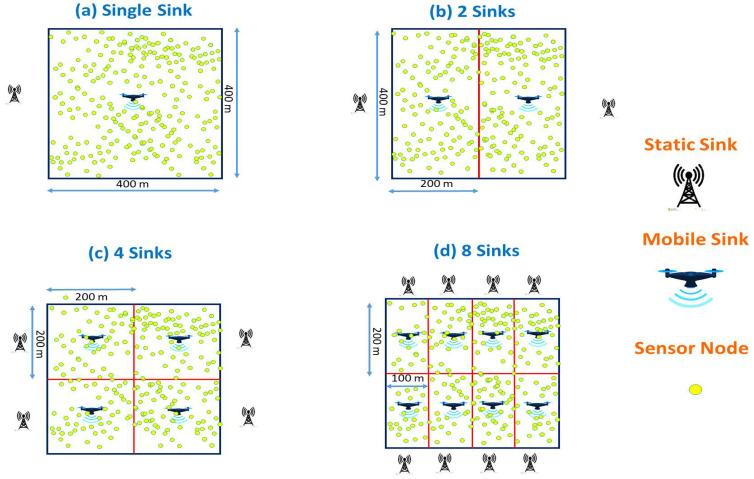
Deployment of static and mobile sinks around the RoI, (**a**) Single Sink for the entire network, (**b**) 2 Sinks for two sub-networks, (**c**) four Sinks for four sub-networks, (**d**) eight Sinks for eight sub-networks.

**Figure 3 sensors-23-08534-f003:**
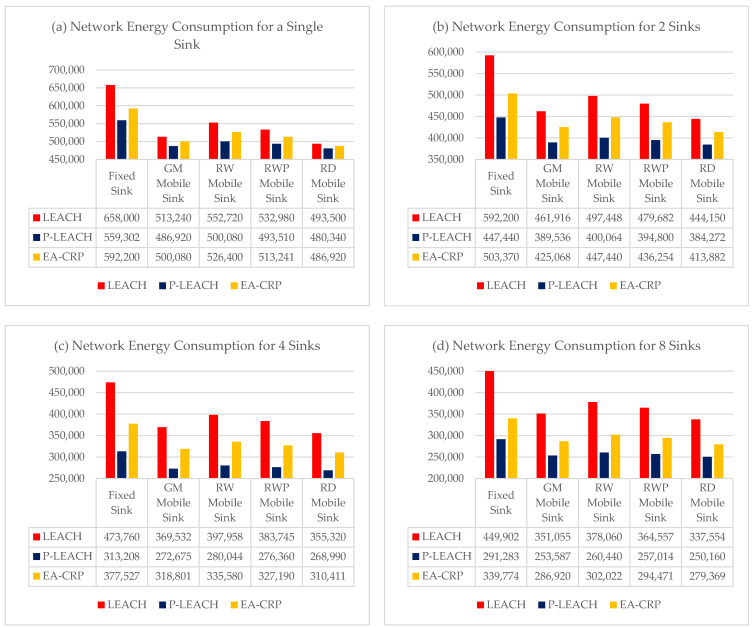
Nodes Energy Consumption with different routing protocols by using multiple statics and mobile sinks, (**a**) Single Sink for the entire network, (**b**) 2 Sinks for 2 sub-networks, (**c**) 4 Sinks for four sub-networks, (**d**) 8 Sinks for 8 sub-networks.

**Figure 4 sensors-23-08534-f004:**
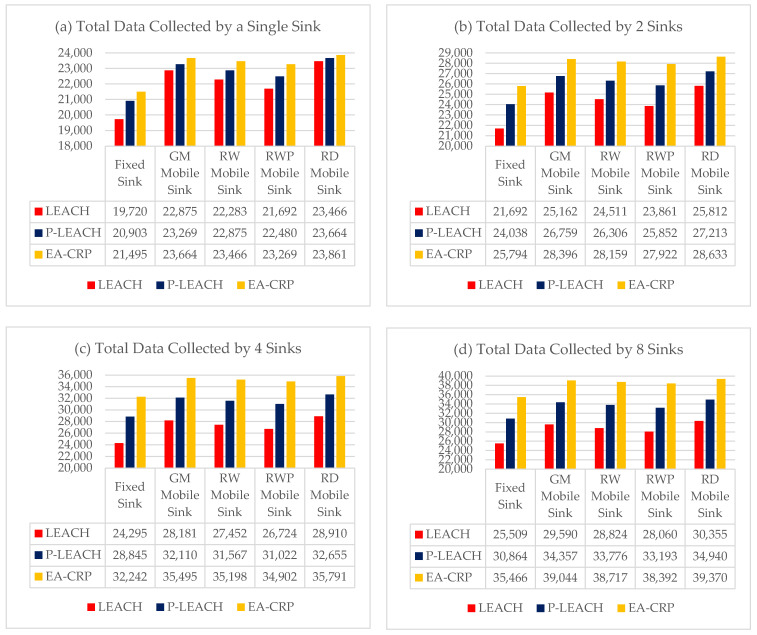
Total data collected by multiple statics and mobiles sinks with different routing protocols, (**a**) Single Sink for the entire network, (**b**) two Sinks for two sub-networks, (**c**) four Sinks for four sub-networks, (**d**) eight Sinks for eight sub-networks.

**Figure 5 sensors-23-08534-f005:**
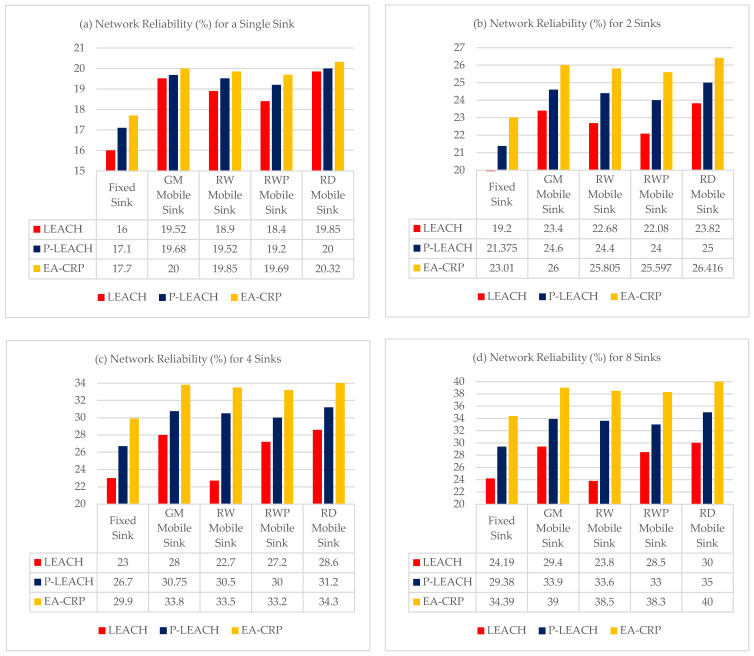
Network Reliability with different routing protocols by using multiple statics and mobiles sinks, (**a**) Single Sink for the entire network, (**b**) two Sinks for two sub-networks, (**c**) four Sinks for four sub-networks, (**d**) eight Sinks for eight sub-networks.

**Figure 6 sensors-23-08534-f006:**
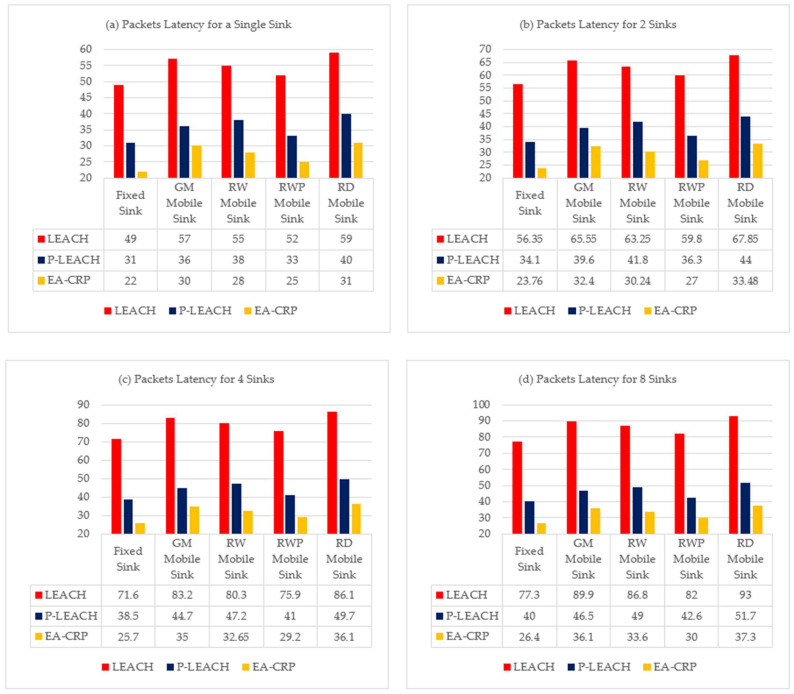
Packets latency with different routing protocols by using multiple statics and mobiles sinks, (**a**) Single Sink for the entire network, (**b**) two Sinks for two sub-networks, (**c**) four Sinks for four sub-networks, (**d**) eight Sinks for eight sub-networks.

**Table 1 sensors-23-08534-t001:** Brief Comparison between cluster-based routing protocols.

Protocol	LEACH (2000)	P-LEACH (2014)	EA-CRP (2018)
Key features	A uniform CH selection process in which every node gets an equal chance for a CH job	The network is divided into partition clusters and uses the prediction techniques with mobile sink tracking	The sensing area is alienated into various layers, and certain clusters are created inside each layer to minimize the communication cost.
Aims	Maximizing the lifetime of the network	Reducing the registration time for new nodes, improving stability, and reducing the energy expenditure of the network.	Reducing the energy expenditure in the network and minimizing the cost of communication between nodes
Strengths	An equal chance is given to every node to become the CH, which divides the workload among all the nodes	The prediction technique helps in minimizing the energy expenditure. The task of the cluster center is handled by the four gateway nodes, which improve the network lifetime	Reduces the cluster setup overhead by dividing the sensing area into layers and sub-layers. The leader heads in different layers reduce the load of CH by performing the data collection and aggregation within the layer. Minimizes the communication cost amid the nodes as the width of the layer decreases towards the BS
Weakness	CH selection is random and does not consider the energy of nodes.	Sink mobility increases the message overhead and complexity	The multilayered structure in the network can cause delays in data transfer.
CH selection method	Select the random number between zero and one and compare it with the threshold to select the CH	The node with the highest battery capacity becomes the cluster center CH	The weight function of energy and distance is calculated for each node. The node with the highest weight function value becomes the CH
Data transmission	Single Hop	Multi-Hop	Multi Hop
CH rotation	YES	NO	YES
Sensor node mobility	Static	Static	Static
topology	Distributed	Distributed	Distributed
Deployment policy	Random	Random	Random

**Table 2 sensors-23-08534-t002:** Parameters of simulations.

Parameter	Value
RoI (Region of Interest)	400 m × 400 m
Number of nodes	800
Number of rounds	1000 rounds
Round Time	20 s
Node deployment (topology)	Random
Packet size	100 bytes
Packet rate	1 packet/s
Initial node energy	100 J
Cluster Routing Protocols	LEACH, P-LEACH and EA-CRP
Sink Mobility models	Static, RWP, RW, RD and GM
Interval of mobility (speed)	[1 m/s–10 m/s]
Time of move	10 s
Time of pause for the RWP model	5 s

**Table 3 sensors-23-08534-t003:** Limitations and Potential Solutions.

Criteria	Limitation	Potential Solutions/Future Research
Random Mobility	The memoryless Random mobility models used in simulations may not accurately represent real-world scenarios where sink mobility can be influenced by various factors such as environmental obstacles (Trees and mountains) or climatic perturbations (wind, rain, and snow)...	Investigate the use of mobility traces collected from real-world deployments to create more accurate random mobility models.
Multiple mobile Sinks Cost	The use of several mobile sinks in simulation can improve the QoS easily and give good results. However, in the real world, there is a significant investment cost behind it.	Consider a hierarchical routing protocol with random mobility awareness of the sinks. The protocol allows dynamic and adaptive coordination between sinks and CHs to optimize routes and reduce the number of mobile sinks and associated costs while respecting important QoS metrics such as delay and coverage.
Optimization of Mobility Patterns	Random mobility models might not adequately represent the movement patterns of mobile sinks. Optimization of the behaviors of these models can be challenging.	Conduct empirical studies to optimize the mobility models with machine learning to ensure they accurately reflect the movement of sinks in realistic WSN applications with QoS constraints.

## Data Availability

Data and codes are available by open request.

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
