# Peer review of "Multiple Mobile Sinks for Quality of Service Improvement in Large-Scale Wireless Sensor Networks"

_sensors, 2023, doi:10.3390/s23208534_

Round 1

Reviewer 1 Report

In the paper, multiple mobile sinks with random mobile models are used to establish a tradeoff between the power consumption and the quality of service.  The proposed routing methods allows minimizing the latency of the transmitted data.

The abstract is well-written and summarizes clearly the content of the paper. It is evident that the contributions of the paper are significant while the topic falls within the scope of the journal.

In the introduction, the most important notions used in the paper such as WSN, QoS, clustering, etc. are briefly mentioned.  The Introduction is generally well-structured  but must be extended to include a more detail description of the main terms and more relevant sources must be included, too. For example, the notion of QoS which is central for the paper is barely mentioned. I recommend to the authors to explain the notion of QoS in more detail and to include references to the documents International Telecommunication Union for example the QoS regulations (ITU-T Supp. 9 of E.800 Series),  the vocabulary for performance, quality of service and quality of experience, etc. Also, the problems related to guaranteeing the QoS in overall telecommunication systems should be discussed.  For example, an overall normalization approach for determining the QoS in overall telecommunication systems is described in:

S. A. Poryazov, et al., "Overall Model Normalization towards Adequate Prediction and Presentation of QoE in Overall Telecommunication Systems," 2019 14th International Conference on Advanced Technologies, Systems and Services in Telecommunications (TELSIKS), Nis, Serbia, 2019, pp. 360-363, doi: 10.1109/TELSIKS46999.2019.9002295.

With regard to the clustering in WSN, more references should be included as this is a hot topic of research. For example, in the recent paper

Han, B.; Ran, F.; Li, J.; Yan, L.; Shen, H.; Li, A. A Novel Adaptive Cluster Based Routing Protocol for Energy-Harvesting Wireless Sensor Networks. Sensors 2022, 22, 1564.

an adaptive hierarchical-clustering-based routing protocol for EH-WSNs (HCEH-UC) is proposed to achieve uninterrupted coverage of the target region through the distributed adjustment of the data transmission.

In Section 2,  a literature review is made. It is excellently written and the cited references are all relevant and up-to-date.  In recent years, various QoS indicators/metrics have been proposed in the literature. Only recently, an important problem has been discussed in the literature,  related to the uncertainty of the service of requests in overall networks. In recent years, intuitionistic fuzzy estimations of the QoS have been proposed. This should be mentioned in the related paragraph of Section 2.

The methodology is excellently described and I have no critical remarks to it.

The numerical results (simulation) are correctly obtained and confirm the reliability and usefulness of the approach.

The conclusions drawn by the authors are supported by the numerical results.

Overall, I evaluate highly the paper and recommend that it be published once the authors address adequately my remarks.

Moderate editting is required. There are some grammar mistakes and some unclear sentences. 

Reviewer 2 Report

The article discusses the increasing involvement of wireless sensor networks (WSNs) in large-scale real-time applications, ranging from hazardous area supervision to military applications. It identifies the challenge of simultaneously improving the quality of service (QoS) and network lifetime in such critical contexts. The proposed solution is to use multiple mobile sinks with random mobility models to address these challenges. The study aims to establish a tradeoff between power consumption and QoS metrics and compares this approach with existing methods that use predefined mobility models for sinks and multi-hop routing techniques.

Strengths:

  1. Novelty and Relevance: The article addresses an important issue in the context of WSNs by introducing the concept of multiple mobile sinks with random mobility models. This approach is novel and highly relevant, especially for real-time applications that demand both efficient energy consumption and high-quality service.

  2. Simulation Results: The article presents simulation results that support the proposed approach's effectiveness. These results demonstrate the potential of hierarchical data routing with random mobile sinks to balance energy distribution among nodes, reduce overall power consumption, minimize latency, increase reliability, and improve data throughput. This empirical evidence strengthens the credibility of the study's claims.

  3. Comprehensive Comparison: The article compares its proposed approach with existing methods that rely on predefined trajectories for mobile sinks and multi-hop architectures. This comparison allows readers to understand the advantages and limitations of the new approach and its potential to outperform established techniques.

Areas for Improvement:

  1. Methodology Explanation: While the article presents simulation results to support its claims, it could benefit from a more detailed explanation of the methodology used for the simulations. Providing a clear and step-by-step description of the simulation setup, parameters, and metrics used for evaluation would enhance the study's transparency and replicability.

  2. Discussion of Limitations: It would be valuable to include a discussion of the limitations of the proposed approach. Real-world implementations may face challenges or constraints that the simulations do not account for. Acknowledging these limitations and discussing potential solutions or areas for future research would provide a more comprehensive perspective.

  3. Citation and References: The article should ensure that it appropriately cites and references related works in the field. This helps readers understand the study's context and the existing body of knowledge.

Conclusion:

The article presents a valuable contribution to the field of wireless sensor networks by introducing a novel approach involving multiple mobile sinks with random mobility models. The simulation results provide compelling evidence that this approach can improve QoS metrics while efficiently managing energy consumption. To enhance the article's quality, the authors should provide a more detailed methodology, address potential limitations, and ensure proper citation and referencing. Overall, this work represents a significant step forward in addressing the challenges of real-time applications in WSNs.

Minor editing of English language required

Round 2

Reviewer 1 Report

I would like to thank the authors for taking into account my critical remarks. I recommend that the paper be published in the present form.

A moderate editting is required. There are unclear sentences and punctuation errors.